# Copy Number Variation Regions Differing in Segregation Patterns Span Different Sets of Genes

**DOI:** 10.3390/ani13142351

**Published:** 2023-07-19

**Authors:** Katherine D. Arias, Juan Pablo Gutiérrez, Iván Fernández, Isabel Álvarez, Félix Goyache

**Affiliations:** 1Área de Genética y Reproducción Animal, SERIDA-Deva, Camino de Rioseco 1225, 33394 Gijón, Spain; kathyah18@gmail.com (K.D.A.); ifernandez@serida.org (I.F.); ialvarez@serida.org (I.Á.); 2Departamento de Producción Animal, Universidad Complutense de Madrid, 28040 Madrid, Spain; gutgar@vet.ucm.es

**Keywords:** CNV segregation, candidate gene, functional analyses, between-individuals differences

## Abstract

**Simple Summary:**

Copy number variations (CNVs) are a substantial fraction of the total genetic variability and have a major effect on phenotypic performance by altering gene expression patterns. However, the identification of CNVs remains challenging. Recent studies suggest that CNVs hardly fit with Mendelian segregation patterns and can be classified as segregating (shared by parents and offspring) and non-segregating. Both CNV classes have been hypothesized to be important in explaining between-populations (segregating CNVs) and between-individuals (non-segregating CNVs) differences in performance. Here, we report that the two CNV classes spanned different sets of genes, thus supporting that theory.

**Abstract:**

Copy number variations regions (CNVRs) can be classified either as segregating, when found in both parents, and offspring, or non-segregating. A total of 65 segregating and 31 non-segregating CNVRs identified in at least 10 individuals within a dense pedigree of the Gochu Asturcelta pig breed was subjected to enrichment and functional annotation analyses to ascertain their functional independence and importance. Enrichment analyses allowed us to annotate 1018 and 351 candidate genes within the bounds of the segregating and non-segregating CNVRs, respectively. The information retrieved suggested that the candidate genes spanned by segregating and non-segregating CNVRs were functionally independent. Functional annotation analyses allowed us to identify nine different significantly enriched functional annotation clusters (ACs) in segregating CNVR candidate genes mainly involved in immunity and regulation of the cell cycle. Up to five significantly enriched ACs, mainly involved in reproduction and meat quality, were identified in non-segregating CNVRs. The current analysis fits with previous reports suggesting that segregating CNVRs would explain performance at the population level, whereas non-segregating CNVRs could explain between-individuals differences in performance.

## 1. Introduction

Copy number variations (CNV) are recognized as markers of inter-individual differences at both the genomic and phenotypic levels [1,2]. They often cover functional DNA sequences with structural variations such as insertion, deletion, or duplication events [3,4]. Moreover, CNVs are a substantial fraction of the total genetic variability, and their importance in altered gene expression is increasingly being recognized [3,5,6,7]. Evolutionary forces such as purifying selective pressures may be involved in any variation in the inheritance of CNVs that will affect the broad spectrum of genomic sequences and could affect gene expression patterns [1,3]. Furthermore, genes spanned in genomic regions where CNVs are identified (CNVRs) may be more specific in their expression patterns at lower and more variable levels than genes mapped elsewhere [8,9]. Individuals may differ in the number of copies of a particular variant in genotype because of an additive effect [10]. Therefore, these sets of genes may play a role in the adaptability and fitness of an organism in response to external pressures [4,11].

Reliable detection of CNVs can be challenged by noisy data [12,13]. Across species, most CNVs are identified either in a single individual or a few unrelated individuals in a population [9,13]. Although CNV contributes significantly to genetic variation [8], the ascertainment of which CNVs are important to explain genetic and phenotypic differences at either the individual or population level is still an issue. Family-based studies allow the classification of CNVRs identified as segregating (those present in the parents and the offspring) and non-segregating (the so-called de novo CNVRs; [14]) CNVRs [3,15]. However, classical pedigree-based analyses of Mendelian inheritance have difficulties in addressing their functionality. Samarakoon et al. [14], in *Plasmodium falciparum*, and Keel et al. [16], in pigs, previously reported significant deviations in Mendelian inheritance patterns in CNVRs. Arias et al. [13] have recently proposed the assessment of the accordance of CNV variation with Mendelian inheritance using pedigree analyses to differentiate these CNVR sets. They suggested that segregating CNVRs are likely to represent “true” genomic variations of importance in a given population whereas non-segregating CNVRs may be important to explain between-individual differences.

This research aimed at filling a scientific gap in genomic research by contributing to the ascertainment of the relationships between different classes of CNVRs and the main physiological processes involved in phenotypic performance. Coding genes spanned in different classes of CNVRs differing in segregation patterns (segregating and non-segregating) were identified in a complex pedigree of the Gochu Asturcelta pig breed. Enrichment analyses were carried out on both sets of genomic regions and the candidate genes spanned were compared. The importance of the different sets of genes to explain either individual or population differences in performance will be discussed to provide new insights into the role of the different classes of CNVR in the performance of livestock populations.

## 2. Materials and Methods

The Gochu Asturcelta pig breed belongs to the Celtic pig strain bred in the Iberian Peninsula [17]. The Celtic-Iberian pigs, which are hypothesized to result from an ancient process of migration of Northern-Central European pigs, with population replacement, into the Iberian Peninsula, were in the majority in Spain and Portugal until the 1950s [17,18]. However, Celtic-Iberian pig breeds, mainly characterized by long, non-compact bodies, good skeletal development, and typically huge ears dropping to the sides of a well-developed head [19], became nearly extinct at the end of the 20th century due to their absorption into and substitution with improved foreign breeds [17]. In 2002, a group of enthusiastic farmers founded the breeders’ association ACGA and initiated a conservation program for the Gochu Asturcelta using six founders showing accordance with the ancient type of the breed, with four of them giving viable offspring only [20,21]. At present, the breed is officially included in the Spanish Catalogue of Livestock Breeds (Regulation APA/53/2007). Gochu Asturcelta individuals are rustic-prolific pigs with slow growth but are well adapted to specific local environments and to using locally available feedstuffs. The characterization of carcass yields and meat quality while keeping traditional extensive or semi-extensive rearing systems is initiated only [22,23]. After the foundation of the breeding program, a strict mating policy was implemented aiming at keeping founder contributions balanced across generations by prolonging the reproductive career of the founders and their direct descendants for as long as possible [17]. This allowed for the construction of a dense pedigree useful for testing different genetic hypotheses across parent–offspring trios and within full-sib litters [13,17,24,25].

### 2.1. Samples and Genotyping

The Gochu Asturcelta pig pedigree included individuals born from 1999 to 2009 (including three founders) and sampled in 14 different farms included in the breeding program of the breeders’ association ACGA. This research used the CNVRs identified in Arias et al. [13], attached in Appendix A. Briefly, the identification of segregating and non-segregating CNVRs was as follows: (a) 492 individuals forming 478 parent-offspring-trios were genotyped using the Axiom_PigHDv1 Array (Axiom_PigHDv1; 545,364 SNPs retained after quality control performed following Arias et al. [25]; and (b) potential CNVRs were identified using two different platforms, PennCNV [26] and QuantiSNP [27]. Finally, candidate CNVRs were constructed using potential CNVRs identified within each platform using the *intersectBed* function of the BedTools version 2.28.0 software [28].

Arias et al. [13] classified CNVRs as segregating CNVRs (69) and non-segregating CNVRs (275). In this survey, CNVRs identified in more than ten individuals were further processed through enrichment analysis to investigate their general importance.

### 2.2. Enrichment and Functional Annotation Analyses

A total of 65 segregating CNVRs and 31 non-segregating CNVRs was subjected to enrichment analyses using the BioMart tool [29] (Appendix A). Protein-coding genes found within these CNVRs were retrieved from the Ensembl Genes 91 database, based on the Sscrofa v11.1 porcine reference genome. All the genes in the genomic areas spanned by segregating and non-segregating CNVRs were processed using the functional annotation tool implemented in DAVID Bioinformatics resources 6.8 [30] to determine enriched functional terms. An enrichment score of 1.3, which is equivalent to the Fisher exact test *p*-value of 0.05 [30], was used as a threshold to define the significantly enriched functional terms in comparison with the whole porcine reference genome background. Selection of significant composite annotation terms (clusters) using the enrichment score as a criterion for selection rather than single annotation terms as independent statistically significant entities supports the identification of biological functions. In other words, moving the analysis of the biological function from the level of single genes to that of biological processes [30,31] allowed us to investigate relationships between GO annotation terms. Relationships among genomic features in different chromosome positions were represented using the software package shinyCircos V2.0 [32].

## 3. Results

The 65 segregating and 31 non-segregating CNVRs subjected to enrichment analyses are described in Appendix A, respectively, and summarized in Table 1. Segregating CNVRs covered about 30 Mb of the genome and were present in all porcine autosomes except for SSC17 and SSC18. Non-segregating CNVRs covered about 14 Mb of the porcine genome and were absent from SSC4, SSC9, and SSC18. Most segregating CNVRs (81% of them) were identified in fewer than 50 individuals (out of 492). Only 3 segregating CNVRs (4.6%) were identified in at least 119 individuals. Non-segregating CNVRs were identified in a smaller number of individuals: 93% of them were identified in fewer than 25 individuals and only 2 CNVRs were identified in more than 35 individuals.

### 3.1. Annotation and Enrichment Analyses in Segregating CNVRs

Enrichment analyses allowed us to annotate 1018 candidate genes within the bounds of the 65 segregating CNVRs selected. A full description of these candidate genes, including their identification, description, and location, retrieved from the Ensembl Genes 91 database, is given in Appendix A and summarized in Table 1.

Functional annotation conducted on genes located on segregating CNVRs allowed us to identify 55 different functional annotation clusters (ACs) (Appendix A), 9 of them being significantly enriched (enrichment score higher than 1.3). These 9 ACs included a total of 45 different genes (Figure 1): ACs1 (enrichment score = 2.20) consisted of 15 trypsin digestion family genes. ACs2 (enrichment score = 1.96) included 4 different genes spanning a chromosomal area of 32.17 kb (from position 127,152 to position 159,320) on SSC2 and involved in immune interferon activity. ACs3 (enrichment score = 1.93) included three genes located on SSC3 and involved in metabolic adaptation, the detoxification of chemotherapeutic drugs, and the catalysis of glutathione. ACs4 (enrichment score = 1.70) included four genes, involved in hemoglobin and oxygen transport activity, spanning a chromosomal area of 13.85 kb (from position 41,478,270 to position 41,492,120) on SSC3. ACs5 (enrichment score = 1.69) involved four genes involved in the catalysis of D-lactic acid, metabolism, and ATP pathways, three of them also identified in ACs3. ACs6 (enrichment score = 1.67) was formed by four genes involved in zinc-binding activity. ACs7 (enrichment score =1.53) included three genes involved in the lipid biosynthetic process and mitochondrial metabolism. ACs8 (enrichment score = 1.46) included three genes of the DNA-binding SAND domain involved in fatty acid metabolism. Finally, ACs9 (enrichment score = 1.42) was formed by eight genes involved in kinase-dependent and kinase-independent catalytic functions.

### 3.2. Annotation and Enrichment Analyses in Non-Segregating CNVRs

Enrichment analyses allowed us to annotate 351 candidate genes in the bounds of the 31 non-segregating CNVRs selected. A full description of these candidate genes, including their identification, description, and location, retrieved from the Ensembl Genes 91 database, is given in Appendix A and summarized in Table 1.

Functional annotation conducted on genes located on segregating CNVRs allowed us to identify 15 different functional annotation clusters (ACns) (Appendix A), 5 of them being significantly enriched (enrichment score higher than 1.3). The 5 functional annotation clusters (ACns) included 29 genes (Figure 2): ACn1 (enrichment score = 3.35) consisted of four ALOX-lipoxygenase family genes located on SSC12 involved in metabolism and nutrient sensing; ACn2 (enrichment score = 2.70) included five different genes spanning a chromosomal area of 239.1 kb (from position 66,191,841 to position 66,430,905) on SSC8 and involved in the metabolism and digestion of bile acids and steroid hormones and liver enzymes; ACn3 (enrichment score = 2.11) included three ALOX-lipoxygenase genes, forming part of ACn1 as well, spanning a chromosomal area of 59.94 kb (from position 53,271,583 to position 53,331,523) on SSC12; ACn4 (enrichment score = 1.75) included three genes involved in the synthesis of unsaturated fatty acids and the chromatin remodeling complex. Finally, ACn5 (enrichment score = 1.57) included four genes involved in ATP-related pathways.

### 3.3. Functional Independence of the Two Sets of Candidate Genes

To ascertain the functional relationship between the candidate genes identified in the bounds of either segregating or non-segregating CNVRs, they were jointly subjected to DAVID analyses. Although the complex inheritance patterns of CNVRs make it difficult to ascertain a non-independence among such physiological processes, results (Appendix A) informed us that the new significant functional clusters identified did not include genes belonging to both sets, therefore suggesting their functional independence.

## 4. Discussion

Copy number variations may act as potentially distal regulators of gene expression by altering regulatory and other functional elements at the transcript or protein level [2,7]. Many works focus on the relationship between CNVRs and production traits [5,33,34,35,36]. CNVR-based studies in pigs showed that candidate genes spanned by copy number alterations have a wide spectrum of molecular functions. Gene ontology analysis of CNVRs suggests that these are mainly enriched in olfactory and sensory perception activities as well as signaling pathways and immunity [5,11]. However, they could also mainly relate to meat quality traits (carcass length, backfat thickness, abdominal fat weight, intermuscular fat content). Thus “olfactory” and “signaling” (under the immunity pathway) appear to be minor representations when compared to the CNV genes in pathways that relate to the meat composition [11]. This could be an enrichment of CNVs toward “environmental sensor” genes (i.e., genes that help to perceive and interact successfully with our ever-changing environment; [3]). The gradual adaptation to an environment and other molecular mechanisms could be influenced by fast-evolving regions in the genome; thus, sensory perception family genes could be involved in the adaptation of breeds to environmental changes [11].

The genomic scenario of Gochu Asturcelta partially departs from this expectation. While genes involved in signaling-related pathways and immunity formed significantly enriched clusters in segregating CNVRs, candidate genes identified within the bounds of non-segregating CNVRs are mainly involved in reproduction and meat quality traits.

The olfactory receptor gene family is one of the largest gene families in the porcine genome [5,16,37], playing an important role in porcine evolution [11]. In fact, olfactory (*OR10A7*, *OR6C2*, *OR8B12*, *OR1G1*, *OR5C1*, *OR13G1*, *OR1D5*, *OR2B11*, *OR2G3*, *OR2G2*, *OR2C3*, *OR5C1*) and taste (*TAS1R3*) receptors are included in the candidate genes spanned into the segregating as well as non-segregating CNVRs identified (Appendix A), but they did not form significantly enriched functional clusters.

Furthermore, the current analysis suggests that the candidate genes spanning segregating and non-segregating CNVRs are functionally independent.

### 4.1. Segregating CNVRs

The significantly enriched functional clusters of the segregating CNVRs identified in the Gochu Asturcelta pig are involved in the immunity and regulation of the cell cycle.

The immunity-related functional clusters are ACs1, ACs2, and ACs9. Serine protease genes (ACs1) are known to be involved in biological processes linked to health and disease in mammals with functions in blood coagulation, fibrinolysis, and immunity [38]. Those genes encode host receptors of viral spike proteins, and their study has been proposed as a way to control viral infections in pigs [39,40]. ACs2 is enriched for GO terms relevant to interferon-mediated signaling pathways. Interferons are involved in immune response pathways including virus recognition, cytokine-mediated signaling pathways, host defense, and innate immunity [41,42]. Finally, genes forming ACs9 are involved in the physiological path of diacylglycerol kinases and the immune function via the T-cell effector promoting the lysis of the target cells and secretion of IFNγ [43]. In this respect, genes belonging to this cluster have been reported to be involved in the immune response (*VAV2* gene; [44]) or antibody response (*PLEKHM1* gene; [45]) in pigs, and or the regulation of the endothelial barrier function (*ARHGAP45* gene; [46]).

Two functional clusters (ACs3 and ACs5) share genes involved in the regulation of different cell functions. In fact, all genes identified in ACs3 are in ACs5. These clusters include genes involved in different gametic processes: the *MEIOB* gene codes a meiotic chromatin-associated protein with an essential function in the meiosis and chromosomal crossover [47], and the *INTS11* gene encodes an endonuclease critical to the transcription and processing of small nuclear RNAs [48]. Furthermore, there exists evidence suggesting that *HAGH* is involved in oxidative stress and that the *HAGHL* gene is up-regulated in the sow genital tract after cervical deposition of the whole boar ejaculate [49,50].

The genes included in ACs4, involved in hemoglobin and oxygen transport activity, are well known to be of importance in livestock populations adapted to high altitudes such as the Tibetan pig [51] or Bos grunniens (Yak; [52]). However, this is not the case for the Gochu Asturcelta breed, a rustic pig traditionally managed in extensive conditions in the hilly country of Asturias (Northern Spain). More likely, the importance of that functional cluster in the Gochu Asturcelta pig population may be related to the genetic relationship between hemoglobin levels and both the sows’ prolificacy and piglets’ survival [53].

ACs6 included genes involved in zinc-binding activity: *TMEM276* and *ZFTRAF1* are genes predicted to enable zinc-ion-binding activity; *TRAF7* and *RNF151* contain an adjacent zinc finger domain involved in the differentiation of muscle tissue [54] and acrosome formation [55], respectively.

The lipid biosynthesis-related clusters are ACs7 and ACs8. Galactose genes are involved in the biosynthesis of meat fatty acids in cattle (*GAL3ST3*; [56]) and intestinal mucin sulfation (*GAL3ST2*; [57]). Moreover, *D2HGDH* is known to stimulate protein synthesis and inhibit protein degradation in muscles [58]. ACs8 included four gene-coding nuclear proteins (SAND domain) with a function in DNA transcriptional control [59]. The genes belonging to the SAND domain are involved in very different cellular processes such as the regulation of the meat-to-fat ratio and carcass quality in pigs (*DEAF1* gene; [35,60]), fatty acids composition and meat color in sheep and early growth stages in chicken (*SAMD11* gene; [61,62]) and the regulation of organ-specific antigen expression (*AIRE* gene; [63]).

### 4.2. Non-Segregating CNVRs

The five significantly enriched clusters identified in non-segregating CNVRs are mainly involved in reproduction and meat quality.

The reproduction-related functional clusters are ACn1, ACn2, ACn3, and ACn5. Lipoxygenase genes included in ACn1 and ACn3 are known to be involved in the formation of the epidermal water barrier. In this respect, these genes have been reported to be involved in the heat stress-induced apoptosis of Sertoli cells in porcine testes [64]. Glucorosyltransferase-genes (ACn2) are known to be involved in the metabolism of steroid hormones: *UGT2* upregulates estrogen signaling and metabolism in the endometrium during the estrous cycle [65] and *LOC100515741* is involved in the hepatic androstenone metabolism contributing to boar taint [66]. Finally, genes forming ACn5 are involved in ATP-related pathways. In this respect, genes belonging to this cluster have been reported to be involved in the hyperactive motility of porcine sperms strongly related to litter size in pigs (*RADIL* gene; [67]); these pathways regulate the carcass adipose tissue under insulin-resistant conditions (*DVL2* gene; [68]).

ACn4 included genes involved in pig meat quality. Fatty acid desaturase genes (*FADS2*, *FADS3*) explain the variations in the monounsaturated and polyunsaturated fatty acid content in the backfat of pigs [69]. Furthermore, the chromatin remodeling complex interacts with external environmental factors in the gene expression of specific phenotypes [11]. In this respect, the *CHD3* gene is reported to be related to the wrinkled skin of the Xiang pig, specifically aging [70].

## 5. Conclusions

The ascertainment of segregation patterns of CNVRs has been suggested to be informative in their use as markers for association analyses with segregating CNVRs probably characterizing performance at a population level. The current analysis adds to this issue and informs us that segregating and non-segregating CNVRs span completely different sets of genes with independent functionality. Previous reports suggesting that segregating CNVRs would explain performance at the population level whereas non-segregating CNVRs are likely to explain differences in performance between individuals are now confirmed. The Gochu Asturcelta pig breed is a non-improved population traditionally managed under semi-extensive conditions. This is supported by the involvement of candidate genes that interact with environmental changes. The breed is considered a reservoir of genes involved in rusticity giving a genomic basis for pig adaptation to a changing environment. The current results will contribute to the conservation of a valuable genetic stock by opening work lines aimed at understanding the Gochu Asturcelta’s ability to perform in harsh environments.

## Figures and Tables

**Figure 1 animals-13-02351-f001:**
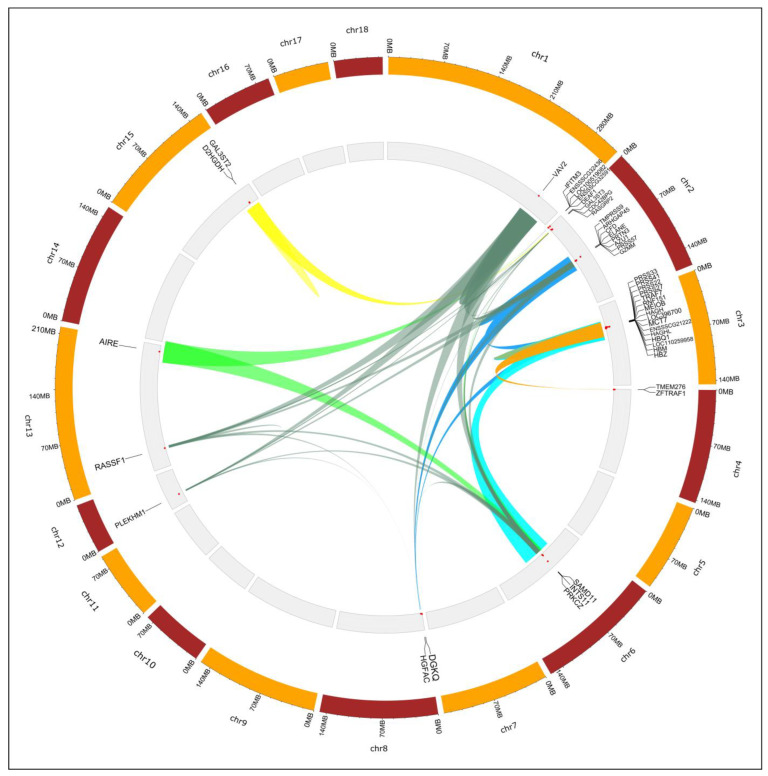
Circular map summarizing information on significantly enriched functional clusters ascertained using candidate genes spanned in segregating CNVRs identified in Gochu Asturcelta pig. The genomic localization of the candidate genes forming the functional clusters is indicated with the gene name or the Gene Stable ID retrieved from the Ensembl Genes 91 database (see Appendix A). At the center of the map, links among candidate genes belonging to the same functional cluster are illustrated using the same color: ACs1, in blue; ACs2, as a thin grey line at the start of SSC2; ACs3 and ACs5, sharing genes on SSC3, in light blue; ACs4 in green on SSC3; ACs6 in orange; ACs7 in yellow; ACs8 in light green; and ACs9 in khaki.

**Figure 2 animals-13-02351-f002:**
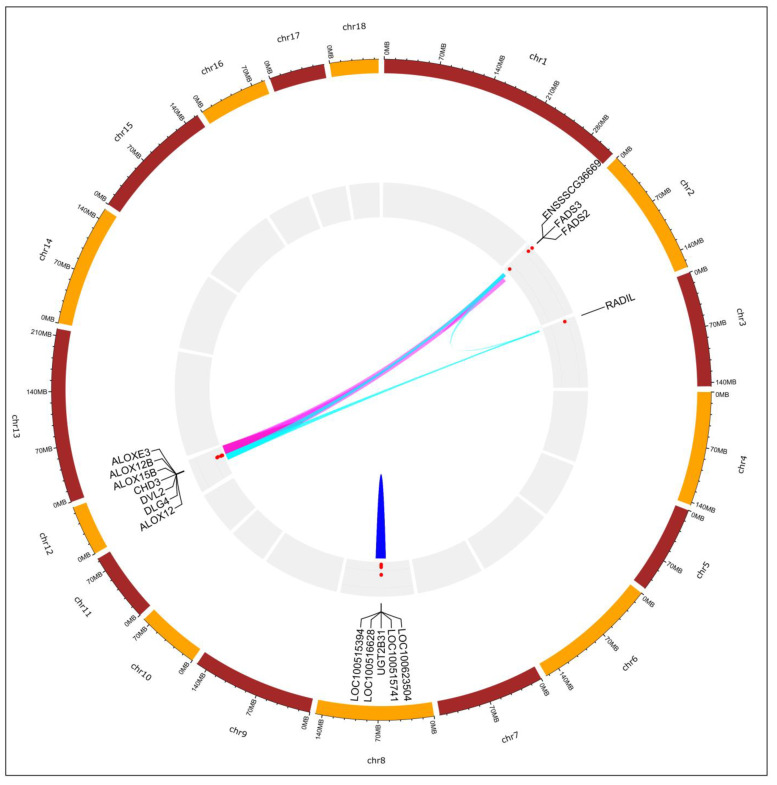
Circular map summarizing information on significantly enriched functional clusters ascertained using candidate genes spanned in non-segregating CNVRs identified in Gochu Asturcelta pig. The genomic localization of the candidate genes forming the functional clusters is indicated with the gene name or the Gene Stable ID retrieved from the Ensembl Genes 91 database (see Appendix A). At the center of the map, links among candidate genes belonging to the same functional cluster are illustrated using the same color: ACn2 in blue on SSC8; ACn4 in pink linking SSC2, and SSC12; ACn5 in light blue, linking genes located on SSC2, SSC3, and SSC12; ACn1 and ACn3, sharing genes located on SSC12 and overlapping with other functional clusters, are not displayed.

**Table 1 animals-13-02351-t001:** Summary of the number of segregating and non-segregating CNVRs identified in Gochu Asturcelta pig and subject to enrichment analyses. The functional pathway for each functional cluster is given in parentheses. The candidate genes assigned to statistically significant (enrichment factor > 1.3) functional clusters identified in segregating (ACs) and non-segregating (ACn) CNVRs are also given. The functional pathway for each functional cluster is given in parentheses.

SSC	Segregating CNVR			Non-Segregating CNVR		
	N	Length (bp)	Candidate Genes	N	Length (bp)	Candidate Genes
1	4	2,233,596	ACs9 (IPR002219): *VAV2*	5	3,341,758	
2	9	7,478,777	ACs1 (IPR001314): *TMPRSS9*, *ELANE*, *CFD*, *PRTN3*, *AZU1*, *PRSS57*, *GZMM* ACs2 (GO:0035455): *IFITM3*, *LOC100519082* ACs7 (IPR009729): *GAL3ST3* ACs8 (IPR000770): *DEAF1* ACs9 (IPR002219): *CDC42BPG*, *RASGRP2*, *ARHGAP45*	6	3,315,384	ACn4 (IPR001199): *FADS3*, *FADS2*
3	3	2,819,269	ACs1 (IPR001314): *PRSS33*, *PRSS41*, *PRSS22*, *PRSS27*, *LOC396700*, *MCT7* ACs3 (IPR017782) and ACs5 (IPR001279): *MEIOB*, *HAGH*, *HAGHL* ACs4 (IPR002338): *HBQ1*, *LOC110259958*, *HBM*, *HBZ* ACs6 (IPR001293): *TRAF7*, *RNF151*, *TMEM276*, *ZFTRAF1*	4	1,882,177	ACn5 (IPR001478): *RADIL*
4	4	1,439,759				
5	2	951,743		1	5808	
6	7	4,075,711	ACs5 (IPR001279): *INTS11* ACs8 (IPR000770): *SAMD11* ACs9 (IPR002219): *PRKCZ*	1	923,089	
7	4	588,387		2	110,830	
8	5	1,858,315	ACs1 (IPR001314): *HGFAC* ACs9 (IPR002219): *DGKQ*	1	229,311	ACn2 (GO:0015020): *LOC100623504*, *LOC100515741*, *UGT2B31*, *LOC100516628*, *LOC100515394*
9	5	760,612				
10	2	328,935		1	4055	
11	4	1,620,698		1	309,506	
12	3	2,185,474	ACs9 (IPR002219): *PLEKHM1*	1	1,242,720	ACn1 (IPR013819): *ALOX12* ACn5 (IPR001478): *DLG4*, *DVL2* ACn4 (IPR001199): *CHD3* ACn1 (IPR013819) and ACn3 (GO:0051122): *ALOX15B*, *ALOX12B*, *ALOXE3*
13	5	1,601,679	ACs8 (IPR000770): *AIRE*, *RASSF1*	1	540,602	
14	4	3,69,933	ACs7 (IPR009729): *D2HGDH*, *GAL3ST2*	3	911,512	
15	3	1,143,291		1	122,959	
16	1	721,470		1	622,344	
17				2	480,937	
Total	65	30,177,649		31	14,042,992	

## Data Availability

The dataset used and analyzed during the current study is available from the corresponding author upon reasonable request.

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
