# Peer review of "Copy Number Variation Regions Differing in Segregation Patterns Span Different Sets of Genes"

_animals, 2023, doi:10.3390/ani13142351_

Round 1

Reviewer 1 Report

The authors identified coding genes spanned in CNVR differing in segre- 55 gation patterns (segregating and non-segregating) on a complex pedigree of the Gochu 56 Asturcelta pig breed. However, the research content of this paper is limited and is lack of novelty, which does not meet the requirements of the journal. Therefore, it is not recommended for publication.

Can be improved

Author Response

Review 1 Report Form

Open Review

(x) Extensive editing of English language required

Yes    Can be improved    Must be improved    Not applicable

Does the introduction provide sufficient background and include all relevant references?

( )    (x)    ( )    ( )

Are all the cited references relevant to the research?

( )    (x)    ( )    ( )

Is the research design appropriate?

( )    (x)    ( )    ( )

Are the methods adequately described?

( )    (x)    ( )    ( )

Are the results clearly presented?

( )    (x)    ( )    ( )

Are the conclusions supported by the results?

( )    ( )    (x)    ( )

Comments and Suggestions for Authors

The authors identified coding genes spanned in CNVR differing in segregation patterns (segregating and non-segregating) on a complex pedigree of the Gochu 56 Asturcelta pig breed. However, the research content of this paper is limited and is lack of novelty, which does not meet the requirements of the journal. Therefore, it is not recommended for publication.

ANSWER: We are sorry. Probably, the novelty of our research was not highlighted enough. The last paragraph of the Introduction section has been modified in depth to avoid this problem. Furthermore, in the current version of our manuscript English has been edited according to the suggestions of the other Reviewers and further revised by a bilingual colleague. We hope the new version of our manuscript reaches the standards of Animals.

Comments on the Quality of English Language

Can be improved

Submission Date

26 May 2023

Date of this review

07 Jun 2023 11:55:40

Reviewer 2 Report

Firstly, I'd like to thank the authors for the great work and bringing to the light the importance of CNV on gene enrichment analysis.  I would like to see this manuscript to be published, but in order to do so additional information needs to be provided and a few conceptual things addressed in the discussion & conclusions:

INTRO

The paper would benefit from a paragraph outlining the breed history of Gochu Asturcelta pig. Please also provide a brief info on typical husbandary conditions, diet and regional distribution (ie, is it widespread globally or constricted to one region) and link this info to your samples. 

1)  492 individuals forming 478 parents-offspring-trios-provide info table on supplementary of ID/origin farm/conditions/DNA sample source 

METHODS

While the methodology and the results seem sound, it is impossible to check these due to the lack of the accessibility of raw data and bioinformatics/analysis parameter sets. 

For the manuscript to be published please ensure as a minimum the following info is publicly accessible (e.g. GitHub, Dryad, Public Google drive etc)

1) Axiom_PigHDv1- provide raw allele call file;

2) PennCNV, QuantiSNP - provide input/ output files and parameter specification

3) intersectBed / BedTools -provide the command script, options and parameters

DISCUSSION

While I agree with the majority of the authors' conclusions assessing your pathway results, I believe there are some that are flawed and should be revised. Specifically, on the functionality of the genes within segregated/ non-segregated regions, which I believe by and large tie in with the breed history and therefore should be appropriately discussed. 

Firstly, authors wrongly interpret the CNVR analysis results of the references 5 & 27 (Line 177-179)  stating the following: "CNVR-based studies in pig showed that candidate genes spanned by Copy Number alterations are enriched in olfactory and sensory perception activity as well as signaling pathways and immunity [5,27]."

After reading these papers what they state is: indeed some genes are involved in sensory (so olfactory could be one of them) phenotypes, but they are mainly  related to "carcass length, backfat thickness, abdominal fat weight, length of scapular, intermuscle fat content of logissimus muscle, body weight at 240 day, glycolytic potential of logissimus muscle, mean corpuscular hemoglobin, mean corpuscular volume and humerus diameter, meat quality traits, back fat thickness and immunity". Thus "olfactory" and "signalling" (under the immunity pathway) appear to be a minor representation when compared to the CNV genes in pathways that relate to the meat composition.  Moreover, one of these studies also included wild bore, thus these adaptations/ trait evolution should be discussed in the light of domestication, heard density, diet, medication and husbandry conditions. 

This is exactly what I believe your results are showing. Although the individual genes between segregated/non-segregated CNVRs are different, I strongly disagree that the biological pathway functionality is independent. Majority of the genes in both groups seem to be involved in energy metabolism, nutrient sensing and conversion, which are likely resulted as a result of selective breeding in order to produce certain meet quality/composition and probably responding to antibiotics/steroids/antinflamatory drugs (speculative based on ACS3 enrichment results- the catalysis of glutathione pathway). In addition this could reflect the switch to higher density populations typical for farmed animals. Although non-segregated CNVR pathways largely fall in the same functional categories (mostly metabolism), there are nuances, which I believe are related to intra- individual lineage husbandry conditions, such as specific diet and mobility (See comments in 3.1.). This is impossible to discern without a better background and sample information, which is why it was requested. 

PRESENTATION

The paper would benefit from minor improvements in the display items. 
1) I'd find easier if the thousands would be comma separated in the SNP numbers

2) Would it be possible to also add the respective pathway for each of the clusters in table 1. 

3) Some fonts in Figure 1 need to be made bigger to be able to read them. 

Author Response

Review 2 Report Form

Quality of English Language

(x) English language fine. No issues detected

Yes    Can be improved    Must be improved    Not applicable

Does the introduction provide sufficient background and include all relevant references?

( )    ( )    (x)    ( )

Are all the cited references relevant to the research?

( )    ( )    (x)    ( )

Is the research design appropriate?

(x)    ( )    ( )    ( )

Are the methods adequately described?

( )    ( )    (x)    ( )

Are the results clearly presented?

( )    (x)    ( )    ( )

Are the conclusions supported by the results?

( )    ( )    (x)    ( )

Comments and Suggestions for Authors

Firstly, I'd like to thank the authors for the great work and bringing to the light the importance of CNV on gene enrichment analysis.  I would like to see this manuscript to be published, but in order to do so additional information needs to be provided and a few conceptual things addressed in the discussion & conclusions:

ANSWER: Thank you for your words. We hope the new version of our manuscript reaches your requirements and the standards of Animals.

INTRO

The paper would benefit from a paragraph outlining the breed history of GochuAsturcelta pig. Please also provide a brief info on typical husbandary conditions, diet and regional distribution (ie, is it widespread globally or constricted to one region) and link this info to your samples.

ANSWER: Thank you. The history and main characteristics of the GochuAsturcelta pig breed is now presented in a new subsection within the Materials and Methods section.

1) 492 individuals forming 478 parents-offspring-trios-provide info table on supplementary of ID/origin farm/conditions/DNA sample source

ANSWER: The available pedigree was previously used in Arias et al. [13], referenced in the text. We add some information about the year of birth and the number of farms sampled (L87-L88). Since that paper is freely available, and intending to avoid including reiterative information in the current paper, we respectfully consider it unnecessary to include more information in the Supplementary files.

METHODS

While the methodology and the results seem sound, it is impossible to check these due to the lack of the accessibility of raw data and bioinformatics/analysis parameter sets.

For the manuscript to be published please ensure as a minimum the following info is publicly accessible (e.g. GitHub, Dryad, Public Google drive etc)

1) Axiom_PigHDv1- provide raw allele call file;

2) PennCNV, QuantiSNP - provide input/ output files and parameter specification

3) intersectBed / BedTools -provide the command script, options and parameters

ANSWER: The analyzed data are part of the PhD Thesis of Miss Katherine Arias which is currently on course. Therefore, we cannot make our data set freely available to avoid parallel research that could be in conflict with the investigations we scheduled. Having said this, the authors are open to discussing any collaboration on non-planned research and the array data may be available from the authors on reasonable request.

DISCUSSION

While I agree with the majority of the authors' conclusions assessing your pathway results, I believe there are some that are flawed and should be revised. Specifically, on the functionality of the genes within segregated/ non-segregated regions, which I believe by and large tie in with the breed history and therefore should be appropriately discussed.

Firstly, authors wrongly interpret the CNVR analysis results of the references 5 & 27 (Line 177-179)  stating the following: "CNVR-based studies in pig showed that candidate genes spanned by Copy Number alterations are enriched in olfactory and sensory perception activity as well as signaling pathways and immunity [5,27]."

After reading these papers what they state is: indeed some genes are involved in sensory (so olfactory could be one of them) phenotypes, but they are mainly related to "carcass length, backfat thickness, abdominal fat weight, length of scapular, intermuscle fat content of logissimus muscle, body weight at 240 day, glycolytic potential of logissimus muscle, mean corpuscular hemoglobin, mean corpuscular volume and humerus diameter, meat quality traits, back fat thickness and immunity". Thus "olfactory" and "signalling" (under the immunity pathway) appear to be a minor representation when compared to the CNV genes in pathways that relate to the meat composition. Moreover, one of these studies also included wild bore, thus these adaptations/ trait evolution should be discussed in the light of domestication, heard density, diet, medication and husbandry conditions.

ANSWER: Thanks for your comments. We agree with the referee’s comment. CNV regions spanned genes involved in many physiological processes mainly related to olfactory and sensory perception. In fact, these primary functions could be related to other physiological functions such as reproduction or meat quality as well. Although the complex inheritance patterns of CNVR make it difficult to ascertain a non-independence among such physiological processes (L170-L173), our research gives new information on the possibility of CNVR differing in segregation patterns can span different sets of genes. This novel information may be the basis for further research refining the knowledge on the issues commented by Reviewer 2.

This is exactly what I believe your results are showing. Although the individual genes between segregated/non-segregated CNVRs are different, I strongly disagree that the biological pathway functionality is independent. Majority of the genes in both groups seem to be involved in energy metabolism, nutrient sensing and conversion, which are likely resulted as a result of selective breeding in order to produce certain meet quality/composition and probably responding to antibiotics/steroids/antinflamatory drugs (speculative based on ACS3 enrichment results- the catalysis of glutathione pathway). In addition this could reflect the switch to higher density populations typical for farmed animals. Although non-segregated CNVR pathways largely fall in the same functional categories (mostly metabolism), there are nuances, which I believe are related to intra- individual lineage husbandry conditions, such as specific diet and mobility (See comments in 3.1.). This is impossible to discern without a better background and sample information, which is why it was requested.

ANSWER: Thanks for your comment. We agree with Reviewer 2: genes spanned by the two CNVR classes analyzed can have similar overall functions (e.g. immunity). However, this is compatible with the fact that they do not participate in the same physiological pathways within those overall functions which are of extreme complexity. To evaluate the independence of functional clusters, both sets of genes were jointly subject to DAVID analyses. The functional clusters identified did not gather genes identified in either segregating or non-segregating CNVR sets. This, therefore, suggests that they may be functionally independent. This has been explicitly stated in the subsection 3.3 of the Results section (L167-L173). Furthermore, a brief background on the Gochu Asturcelta breed is now given in the Materials and Methods section (L65-L85). We hope that our results can now be better understood.

PRESENTATION

The paper would benefit from minor improvements in the display items.

1) I'd find easier if the thousands would be comma separated in the SNP numbers

ANSWER: commas are now separating thousands in all cases.

2) Would it be possible to also add the respective pathway for each of the clusters in table 1.

ANSWER: done as suggested.

3) Some fonts in Figure 1 need to be made bigger to be able to read them.

ANSWER: improved, as suggested

Submission Date

26 May 2023

Date of this review

15 Jun 2023 22:15:10

Reviewer 3 Report

Dear authors,

I have read and reviewed the manuscript entitled “Copy Number Variation Regions differing in segregation patterns span different sets of genes.”. The article deals with functional annotation analysis in segregating and non-segregating CNVR. It is part of a work that includes two other published manuscripts. It is a good work, easily readable and generally well-written in structure and concepts. I recommend it for publication with a few minor suggestions to consider.

In your survey, you clearly highlighted that different classes of CNVR include different sets of genes with independent functionality.

At the end of the introduction, you stated “Inferences on the importance of the different sets of genes to explain either individual or population differences in performance will be discussed.” The discussion section does not include a real inference on the role of the different genes which help explain individual or population-level differences.

Then you concluded that these results confirm the usefulness of segregating CNVR for population-level analyses and the usefulness of non-segregating CNVR for explaining individual variability of performances. “Previous reports suggesting that segregating CNVR would explain performance at the population level whereas non-segregating CNVR are likely to explain differences in performance between individuals are now confirmed.” This appears to me a syllogism based on hypotheses (strong) in the scientific literature which are not demonstrated by this work.

I would suggest modifying those phrases focusing on current results. Alternatively, the authors could further discuss the results of the enrichment analysis to bring the different gene sets identified to the population rather than individual issues.

Moreover, in the conclusion section, I would expect the authors to outline what the practical implications of their results could be on the selection and recovery programmes of the Gochu Asturcelta pig breed.

Here are some suggestions to improve the readability of the article.

Line 10: replace “that” with “and”

Line 11: However, the identification…

Line 13: replace "to explain” with “in explaining”

Line 15: Here,

Line 16: replace "therefore” with “thus”

Line 17-18: rewrite as “…segregating when found in both parents and offspring or non-segregating.”

Line 21: “allowed us”

Line 22: what do you mean with “in the bounds”? within? “In the bounds” is repeated through the text.

Line 24: “allowed us”

Line 26: “of the cell cycle”

Line 29: “…level, whereas…”

Line 34: “as markers”

Lin e 35: “cover”

Line 37: replace “in turn” with “moreover”

Line 48-49: “allow the classification of”

Line 55: aimed

Line 55-56: spanned in different classes of CNVR

Line 57-58: I suggest rewriting it as “Enrichment analyses were carried out on both types of genomic regions, and the relative candidate genes were compared.”

Line 59-60: “of genes, explaining individual or population differences in performance, were discussed.”

Line 63: replace “started” with “used”

Line 63-64: “see Supplementary Table S3 and Table S4 of that paper.” I do not like the direct reference to tables of a different manuscript even if it is strictly related to this. I suggest the authors to prepare an original supplementary table or delete the reference.

Line 65: delete “Up to”

Line 70: “using the potential”

Line 72: “Arias et al [12] classified”

Line 73-74: I suggest rewriting it as “In this survey, CNVR identified in more than ten individuals were further processed through the enrichment analysis to investigate their general importance.”

Line 74-76: replace “subject” with “subjected” and delete “only”. In addition, I suggest moving the phrase to the next paragraph.

Line 78-79: rephrase based on the previous suggestion

Line 79: delete “identified”

Line 89-91: I suggest rephrasing as “In other words, moving the analysis of biological function from the level of single genes to that of biological processes [20,21] allowed us to investigate relationships between GO annotation terms.”

Line 95: CNVR subjected to

Line 98: delete “porcine”

Line 99: and were absent in

Line 100: out of

Line 102: of individuals: 93 % of them

Line 114-116: I suggest rewriting as “ACs2 (enrichment score = 1.96) included 4 different genes spanning a chromosomal area of 32.17 kb (from position 127,152 to position 159,320) on SSC2 and involved in interferon activity.”

Line 116: three genes located on SSC3 and involved

Line 120-121: three of them also identified in ACs3

Line 135: delete “selected”

Line 136-139: I suggest rewriting as “ACn2 (enrichment score = 2.70) included 5 different genes spanning a chromosomal area of 239.1 kb (from position 66,191,841 to position 66,430,905) on SSC8 and involved in the metabolism of bile acids and steroid hormones;”

Line 142-143: and the chromatin remodelling complex.

Line 147: were jointly subjected

Line 176-183: I would organize the paragraph this way

“Copy number variations may act as potential distal regulators of gene expression by altering regulatory and other functional elements at the transcript or protein level [2,7].

Although many works focus on the relationship between CNVR and production traits [5,23–26], CNVR-based studies in pig species showed that candidate genes involved in Copy Number alterations are enriched in olfactory and sensory perception activity as well as signalling pathways and immunity [5,27].

This could be an enrichment mechanism toward “environmental sensor” genes (i.e. genes that help to perceive and interact successfully with our everchanging environment; [3]).”

Line 190: candidate genes spanning segregating

Line 193: While candidate genes within the non-segregating

Line 206: Diacylglycerol Kinases and in the immune function via T cell effector promoting

Line 212: In fact, all genes

Line 217: replace “as well as” with “and”

Line 226: both the sows' prolificacy and

Line 254:      ; these pathways also regulate

Line 256: in pig meat quality

Line 259: of the Xiang pig, especially ageing.

The English Language is not an issue despite a few typos in the text that can be easily improved.

Author Response

Review 3 Report Form

Open Review

Quality of English Language

(x) Minor editing of English language required

Yes    Can be improved    Must be improved    Not applicable

Does the introduction provide sufficient background and include all relevant references?

( )    (x)    ( )    ( )

Are all the cited references relevant to the research?

(x)    ( )    ( )    ( )

Is the research design appropriate?

(x)    ( )    ( )    ( )

Are the methods adequately described?

(x)    ( )    ( )    ( )

Are the results clearly presented?

(x)    ( )    ( )    ( )

Are the conclusions supported by the results?

( )    (x)    ( )    ( )

Comments and Suggestions for Authors

Dear authors,

I have read and reviewed the manuscript entitled “Copy Number Variation Regions differing in segregation patterns span different sets of genes.”. The article deals with functional annotation analysis in segregating and non-segregating CNVR. It is part of a work that includes two other published manuscripts. It is a good work, easily readable and generally well-written in structure and concepts. I recommend it for publication with a few minor suggestions to consider.

ANSWER: Thank you for your words. We hope the new version of our manuscript reaches your requirements and the standards of Animals.

In your survey, you clearly highlighted that different classes of CNVR include different sets of genes with independent functionality.

At the end of the introduction, you stated “Inferences on the importance of the different sets of genes to explain either individual or population differences in performance will be discussed.” The discussion section does not include a real inference on the role of the different genes which help explain individual or population-level differences.

Then you concluded that these results confirm the usefulness of segregating CNVR for population-level analyses and the usefulness of non-segregating CNVR for explaining individual variability of performances. “Previous reports suggesting that segregating CNVR would explain performance at the population level whereas non-segregating CNVR are likely to explain differences in performance between individuals are now confirmed.” This appears to me a syllogism based on hypotheses (strong) in the scientific literature which are not demonstrated by this work.

I would suggest modifying those phrases focusing on current results. Alternatively, the authors could further discuss the results of the enrichment analysis to bring the different gene sets identified to the population rather than individual issues.

ANSWER: Thank you for your comments. The inheritance patterns of segregating CNVR could explain their importance at the population level. However, non-segregating CNVR, occurring in multiple progenies but in neither parent, absolutely departed from Mendelian inheritance rules and it is more likely to explain individual differences in performance. Our results suggest a possible functional independency between candidate genes spanned by each class of CNVR. While candidate genes involved in signaling-related and immunity pathways formed significantly enriched clusters in the case f segregating CNVR, candidate genes identified in the bounds of non-segregating CNVR are mainly involved in reproduction and meat quality traits. However, it is not excluded that segregating genes perform as “environmental sensor” genes (Ionita-Laza et al., 2009) involved in the adaptation of breeds to environmental changes (Panda et al., 2022). This is now explicitly stated in the first paragraph of the Discussion section (L198-L215).

Moreover, in the conclusion section, I would expect the authors to outline what the practical implications of their results could be on the selection and recovery programmes of the GochuAsturcelta pig breed.

ANSWER: Thank you for your comments. We have followed your suggestions and modified the Conclusions section accordingly (L298-L304).

Here are some suggestions to improve the readability of the article.

Line 10: replace “that” with “and”

ANSWER: Thank you. Modified as suggested.

Line 11: However, the identification…

ANSWER: Thank you. Modified as suggested.

Line 13: replace "to explain” with “in explaining”

ANSWER: Thank you. Modified as suggested.

Line 15: Here,

ANSWER: Thank you. Modified as suggested.

Line 16: replace "therefore” with “thus”

ANSWER: Thank you. Modified as suggested.

Line 17-18: rewrite as “…segregating when found in both parents and offspring or non-segregating.”

ANSWER: Thank you. Modified as suggested.

Line 21: “allowed us”

ANSWER: Thank you. Modified as suggested.

Line 22: what do you mean with “in the bounds”? within? “In the bounds” is repeated through the text.

ANSWER: We refer to the genomic limits of the CNVR. We have used that term in previous papers.

Line 24: “allowed us”

ANSWER: Thank you. Modified as suggested.

Line 26: “of the cell cycle”

ANSWER:  Thank you. Modified as suggested.

Line 29: “…level, whereas…”

ANSWER: Thank you. Modified as suggested.

Line 34: “as markers”

ANSWER: Thank you. Modified as suggested.

Lin e 35: “cover”

ANSWER: Thank you. Modified as suggested.

Line 37: replace “in turn” with “moreover”

ANSWER: Thank you. Modified as suggested.

Line 48-49: “allow the classification of”

ANSWER: Thank you. Modified as suggested.

Line 55: aimed

ANSWER: Thank you. Modified as suggested.

Line 55-56: spanned in different classes of CNVR

ANSWER: Thank you. Modified as suggested.

Line 57-58: I suggest rewriting it as “Enrichment analyses were carried out on both types of genomic regions, and the relative candidate genes were compared.”

ANSWER: Thank you. Modified as suggested.

Line 59-60: “of genes, explaining individual or population differences in performance, were discussed.”

ANSWER: Thank you. Modified as suggested.

Line 63: replace “started” with “used”

ANSWER: Thank you. Modified as suggested.

Line 63-64: “see Supplementary Table S3 and Table S4 of that paper.” I do not like the direct reference to tables of a different manuscript even if it is strictly related to this. I suggest the authors to prepare an original supplementary table or delete the reference.

ANSWER: Thank you for your suggestion. We prepared an additional supplementary table wich summarized the CNVR previously identified by Arias et al., 2023. (see Supplementary Table S1: List of Copy Number Variation Regions identified by Arias et al. [13] in Gochu Asturcelta pig using the software PennCNV and QuantiSNP.).

Line 65: delete “Up to”

ANSWER: Thank you. Modified as suggested.

Line 70: “using the potential”

ANSWER: Thank you. Modified as suggested.

Line 72: “Arias et al [12] classified”

ANSWER: Thank you. Modified as suggested.

Line 73-74: I suggest rewriting it as “In this survey, CNVR identified in more than ten individuals were further processed through the enrichment analysis to investigate their general importance.”

ANSWER: Thank you. Modified as suggested.

Line 74-76: replace “subject” with “subjected” and delete “only”. In addition, I suggest moving the phrase to the next paragraph.

ANSWER: Thank you. Modified as suggested.

Line 78-79: rephrase based on the previous suggestion

ANSWER: Thank you. Modified as suggested.

Line 79: delete “identified”

ANSWER: Thank you. Modified as suggested.

Line 89-91: I suggest rephrasing as “In other words, moving the analysis of biological function from the level of single genes to that of biological processes [20,21] allowed us to investigate relationships between GO annotation terms.”

ANSWER: Thank you. Modified as suggested.

Line 95: CNVR subjected to

ANSWER: Thank you. Modified as suggested.

Line 98: delete “porcine”

ANSWER: Thank you. Modified as suggested.

Line 99: and were absent in

ANSWER: Thank you. Modified as suggested.

Line 100: out of

ANSWER: Thank you. Modified as suggested.

Line 102: of individuals: 93 % of them

ANSWER: Thank you. Modified as suggested.

Line 114-116: I suggest rewriting as “ACs2 (enrichment score = 1.96) included 4 different genes spanning a chromosomal area of 32.17 kb (from position 127,152 to position 159,320) on SSC2 and involved in interferon activity.”

ANSWER: Thank you. Modified as suggested.

Line 116: three genes located on SSC3 and involved

ANSWER: Thank you. Modified as suggested.

Line 120-121: three of them also identified in ACs3

ANSWER: Thank you. Modified as suggested.

Line 135: delete “selected”

ANSWER: Thank you. Modified as suggested.

Line 136-139: I suggest rewriting as “ACn2 (enrichment score = 2.70) included 5 different genes spanning a chromosomal area of 239.1 kb (from position 66,191,841 to position 66,430,905) on SSC8 and involved in the metabolism of bile acids and steroid hormones;”

ANSWER: Thank you. Modified as suggested.

Line 142-143: and the chromatin remodelling complex.

ANSWER: Thank you. Modified as suggested.

Line 147: were jointly subjected

ANSWER: Thank you. Modified as suggested.

Line 176-183: I would organize the paragraph this way

ANSWER: Thank you. Modified as suggested.

“Copy number variations may act as potential distal regulators of gene expression by altering regulatory and other functional elements at the transcript or protein level [2,7].

ANSWER: Thank you. Modified as suggested.

Although many works focus on the relationship between CNVR and production traits [5,23–26], CNVR-based studies in pig species showed that candidate genes involved in Copy Number alterations are enriched in olfactory and sensory perception activity as well as signalling pathways and immunity [5,27].

ANSWER: Thank you. Modified as suggested.

This could be an enrichment mechanism toward “environmental sensor” genes (i.e. genes that help to perceive and interact successfully with our everchanging environment; [3]).”

ANSWER: Thank you. Modified as suggested.

Line 190: candidate genes spanning segregating

ANSWER: Thank you. Modified as suggested.

Line 193: While candidate genes within the non-segregating

ANSWER: Thank you. Modified as suggested.

Line 206: Diacylglycerol Kinases and in the immune function via T cell effector promoting

ANSWER: Thank you. Modified as suggested.

Line 212: In fact, all genes

ANSWER: Thank you. Modified as suggested.

Line 217: replace “as well as” with “and”

ANSWER: Thank you. Modified as suggested.

Line 226: both the sows' prolificacy and

ANSWER: Thank you. Modified as suggested.

Line 254:      ; these pathways also regulate

ANSWER: Thank you. Modified as suggested.

Line 256: in pig meat quality

ANSWER: Thank you. Modified as suggested.

Line 259: of the Xiang pig, especially ageing.

ANSWER: Thank you. Modified as suggested.

Comments on the Quality of English Language

The English Language is not an issue despite a few typos in the text that can be easily improved.

ANSWER: English has been edited according to your suggestions, those by Reviewer 2, and further revised by a bilingual colleague. We hope the new version of our manuscript reaches your requirements and the standardsof Animals.

Submission Date

26 May 2023

Date of this review

13 Jun 2023 15:25:16

Round 2

Reviewer 1 Report

The authors have made a commendable effort to address most of reviewers’ concerns within the available space. I am satisfied with the revised version of the paper. I recommend published in this journal.

The authors have made a commendable effort to address most of reviewers’ concerns within the available space. I am satisfied with the revised version of the paper. I recommend published in this journal.

Author Response

Thank you for your words.

Following your suggestions we have included more information in the Introduction section to highlight both the interest of the problem faced and the novelty of our research.

Furthermore, a bilingual colleague has revised the paper again and included few improvements of English grammar and style.

To facilitate the reviewing process, changes have been typed in red.

We hope that the new version of our works reaches the standards of Animals.

Best regards

Félix Goyache
